# Protective Effect of Amber Extract on Human Dopaminergic Cells against 6-Hydroxydopamine-Induced Neurotoxicity

**DOI:** 10.3390/molecules27061817

**Published:** 2022-03-10

**Authors:** Yuening Luo, Siqi Zhou, Reiko Takeda, Kazuma Okazaki, Marie Sekita, Kazuichi Sakamoto

**Affiliations:** 1Graduate School of Life and Environmental Sciences, University of Tsukuba, Tsukuba 305-8572, Ibaraki, Japan; shiwasuyayoi@gmail.com (Y.L.); zhousiqi66@gmail.com (S.Z.); r-takeda@yamanobeautychemical.com (R.T.); 2Research and Development, Kohaku Bio Technology Co., Ltd., Tsukuba 305-8572, Ibaraki, Japan; k-okazaki@yamanobeautychemical.com (K.O.); m-sekita@yamanobeautychemical.com (M.S.)

**Keywords:** 6-hydroxydopamine, amber, extracellular signal-regulated kinase pathway, autophagy

## Abstract

Parkinson’s disease (PD) is the second most common progressive neurodegenerative disease, after Alzheimer’s disease. In our previous study, we found that amber—a fossilized plant resin—can protect cells from apoptosis by decreasing the generation of reactive oxygen species (ROS). In this study, we focused on the effect of amber on 6-hydroxydopamine-induced cell apoptosis in the human neuroblastoma cell line SHSY5Y (one model for PD). Initially, we determined the protective effect of amber on the PD model. We found that amber extract has a protective effect against 6-hydroxydopamine-induced cell apoptosis. The decrease in ROS, cleaved caspase-3, pERK, and extracellular signal-regulated kinase (ERK) protein levels confirmed that amber extract decreases apoptosis via the ROS-mediated ERK signaling pathway. Furthermore, we determined the effects of amber extract on autophagy. The results showed that amber extract increased the levels of LC3II and Beclin-1, suggesting that amber extract can protect neuronal cells against 6-hydroxydopamine-induced cell apoptosis by promoting autophagy.

## 1. Introduction

Parkinson’s disease (PD) is the second most common progressive neurodegenerative disease, after Alzheimer’s disease, and is characterized by movement disorders such as resting tremor, bradykinesia, rigidity, and postural instability [1]. The prevalence of PD increases with age, affecting approximately 1% of the population aged over 60 years, and 2–4% of the population aged over 80 years [2,3]. Similar to many other neurological disorders, the cause of PD is still not completely understood. The death of dopaminergic neurons in the substantia nigra pars compacta, striatal dopamine depletion, and the presence of α-synuclein aggregates are the neuropathological hallmarks of PD [4,5]. Mitochondrial dysfunction, neuroinflammation, and oxidative stress have also been reported to be associated with PD [6,7,8]. Recently, α-synuclein aggregation has been identified as a therapeutic target for PD [5]. A previous study reported that α-synuclein has two types of degradation pathways: proteasomal and autophagic proteolysis [9,10]. Unlike unfolded proteins, aggregated proteins are usually resistant to unfolding, and their degradation is highly dependent on autophagy [11]. The autophagy enhancer rapamycin has been shown to have a protective effect on PD-related dopaminergic neurodegeneration [10]. According to a previous study, autophagy, which eliminates aggregated α-synuclein, may represent a potential neuroprotective strategy in PD [9,12].

6-Hydroxydopamine (6-OHDA) is commonly used to develop in vitro models of PD [13]. The structure of 6-OHDA is similar to that of dopamine, and it can easily oxidize into hydrogen peroxide and para-quinone [14]; it can enter neuronal cells and generate intracellular reactive oxygen species (ROS), ultimately causing neuronal cell death [15].

Amber is a type of fossilized plant resin that is commonly used as jewelry and in decorative objects. In China, amber has been used as a medicine for more than a thousand years to enhance mental stability, stop bleeding, aid in wound healing, and act as a diuretic. The medicinal effects of amber had already been mentioned in the book *Lei’s Treatise on the Preparation of Medicinal Substances* (Lei Gong Pao Zhi Lun) around the 5th century CE. In Russia, amber is commonly used in folk medicine [16]. In recent years, the effects of amber on allergy [17], anti-inflammation [18], suppression of melanin production, promotion of collagen production [19], and reduction in fat accumulation [20] have been identified. A previous study by our group found that amber extract has a protective effect against amyloid-β-induced neuronal cell death [21]. In this study, we aimed to investigate the protective effects of amber extracts against 6-OHDA-induced neuronal cell apoptosis.

## 2. Results

### 2.1. Amber Extract Protects SHSY5Y Cells against 6-OHDA-Induced Cytotoxicity

Cell viability was measured using the MTT assay. Low concentrations of amber extract (below 50 μg/mL) showed no cytotoxicity. However, cells treated with 100 μg/mL amber extract showed significantly decreased viability compared with the cells in the control group (Figure 1A).

Compared with the control group, a significant decrease in viability was observed for cells treated with 75 µM 6-OHDA, whereas treatment with amber extract showed a protective effect against 6-OHDA -induced cytotoxicity (Figure 1B).

### 2.2. Amber Extract Decreases 6-OHDA-Induced Apoptosis in SHSY5Y Cells

To evaluate the effect of amber extract on apoptosis, the apoptosis assay was performed, and the protein expression levels of caspase-3 were measured. Cells treated with 75 µM 6-OHDA showed a significant increase in cell death and cleaved caspase-3 protein levels when compared with cells in the control group. In contrast, apoptosis and cleaved caspase-3 protein levels were lower in the amber-extract-treated group than in the 6-OHDA group (Figure 2A–C).

### 2.3. Amber Extract Decreases 6-OHDA-Induced ROS Generation in SHSY5Y Cells

Cells treated with 75 µM 6-OHDA showed a significant increase in ROS levels compared with the cells in the control group; however, the levels decreased with amber extract treatment (Figure 3A).

We also measured pERK protein levels, which are related to ROS generation. The pERK/ERK ratio was increased in the 6-OHDA group compared with that in the control group, and decreased in the amber extract treatment (Figure 3B,C).

### 2.4. Effect of Amber Extract on Autophagy-Related Gene Expression in SHSY5Y Cells

To confirm the effect of amber extract on autophagy, the protein expression levels of autophagy-related genes, Beclin-1, and LC3 were measured. We observed that 6-OHDA decreased the protein levels of Beclin-1, but had no significant effect on LC3-II levels. However, the protein levels of Beclin-1 and LC3-II were higher in the amber-extract-treated group than in the 6-OHDA group (Figure 4A–C). Furthermore, we measured the LC3 II/LC3 I ratio as another marker of autophagy. Cells treated with 6-OHDA showed no significant differences in the LC3 II/LC3 I ratio compared with cells in the control group; however, the LC3 II/LC3 I ratio was increased in the amber extract group (Figure 4D). These results suggest that amber extract promotes autophagy in SHSY5Y cells.

## 3. Discussion

Amber is a type of fossilized resin obtained from plants. At present, the largest known deposit of amber is the Baltic region. Baltic amber is the most popular, and is commonly used in folk medicine in Russia [16]; it contains many bioactive compounds, such as monoterpenes, succinic monoterpenoids, sesquiterpenoids, and other compounds [16,22]. Monoterpenes and monoterpenoids are a series of chemicals that are widely diffused in plants; their basic structure is composed of two bound isoprene units [23]. Monoterpenes have been reported to have antioxidant, anti-inflammatory, antidiabetic, hepatoprotective, and antitumor activities, and can also modulate autophagy [24]. Monoterpenoids also have been reported to have anti-inflammatory effects [25], and as a kind of treatment for chronic pain [26]. Moreover, sesquiterpenoids have also been reported to have anti-inflammatory [27] and neuroprotective effects [28]. Previous studies have shown that Baltic amber contains several different active compounds [16,29], such as p-cymene and dehydroabietic acid, which have been reported to have anti-inflammatory effects [30,31]; camphor, which has been reported to have antifungal activity [32]; pimaric acid, which has been reported to have anti-atherosclerotic activity [33], etc. However, there are still many other unknown components in amber, and future studies should focus on identifying all components of the amber extract and elucidating the exact components that have potential neuroprotective effects.

In previous studies, many effects of amber—such as anti-allergic effects [17], promotion of collagen production, and suppression of melanin production [19]—have been investigated. In our previous studies, we found that amber extract also has anti-inflammatory effect [18], the ability to reduce fat accumulation [20], and can protect cells from amyloid-β1-42-induced cytotoxicity [21].

6-OHDA can be taken up by the dopamine transporter (DAT) into cells and generate active oxygen, which causes the death of neurons [15,34]. In our pervious study, amber extract was able to upregulate the mRNA levels of SOD1 and SOD2 in order to downregulate ROS generation [21]. Therefore, the purpose of this study was to focus on the effects of amber on 6-OHDA-induced generation of active oxygen and neuron death.

In a previous study, differentiated SHSY5Y cells underwent some alteration in the AKT pathway, and had higher tolerance to the toxicity of 6OHDA [35], meaning that undifferentiated SHSY5Y cells may be more suitable for a Parkinson’s disease model. Therefore, in this study, undifferentiated SHSY5Y cells were used.

Cell viability was no different in the amber extract group (up to 50 μg/mL); however, it was significantly increased in the amber pretreatment group compared with that in the 6-OHDA group. Thus, we concluded that amber extract protected the cells from 6-OHDA-induced cytotoxicity. Moreover, this result indicates that amber may have a preventive effect against 6OHDA cytotoxicity.

An apoptosis assay was performed to confirm the effects of amber extract on 6-OHDA-induced cell death. The fluorescence intensity of apoptotic cells to healthy cells was increased in the 6-OHDA group—which is consistent with the findings of a previous study [36]—and decreased in the amber pretreatment group. Moreover, the protein levels of caspase-3 and cleaved caspase-3 were measured. The results of the amber group showed that cleaved caspase-3 levels showed a concentration-dependent decrease compared with those in the 6-OHDA group. This result suggests that amber has a protective effect against 6-OHDA-induced cell death.

In recent years, many studies have suggested that mitochondrial dysfunction [37] and oxidative stress [7,38] play important roles in PD. In previous studies, some compounds—such as succinate [16]—were reported to have the function of reducing mitochondrial dysfunction [39] and recovering mitochondrial oxygen consumption [40]. As succinate is one of the components of amber, this means that amber may also participate in the regulation of mitochondrial function.

6-OHDA can enter neuronal cells and generate intracellular ROS, causing neuronal cell death [15,34]. In our study, the increase in ROS in the 6-OHDA group and the decrease in ROS in the amber group were clearly observed. Therefore, amber reduced cell apoptosis because of the possible effect of ROS scavenging. In a previous study, oxidative stress induced by ROS generation led to apoptosis via caspase-3 activation [38]. However, unlike cleaved caspase-3, ROS generation was not significantly different between the cells treated with various concentrations of amber extract. This suggests that in addition to its ROS-scavenging effect, amber may exert its effect via another signaling pathway to protect cells from apoptosis.

ERK1/2 have been reported as important regulators of neuronal responses [41,42]. Activation of the ERK signaling pathway may be related to increased ROS generation [43] and cell death [41]. In vitro and postmortem studies have shown that ERK1/2 activation plays an important role in 6-OHDA-induced cell death [44,45]. Therefore, we measured the protein levels of pERK and ERK. The results showed that the phosphorylation of ERK in the 6-OHDA group was significantly increased, consistent with a previous study [41], whereas amber inhibited the phosphorylation of ERK. In a previous study, the suppression of the activation of ERK decreased ROS-induced cell death [43]. This also suggests that amber extract decreases apoptosis via the ROS-mediated ERK signaling pathway.

Pervious research has shown that short-term (30 min) treatment of 6OHDA can lead to a distinct temporal pattern in the activation of ERK—first at 15 min, and again from 6 h to 24 h. The activation of ERK at early stages exerts a protective effect on cells, and the inhibition of the first activation leads to the increase in ROS [46] However, the long-term and toxic concentration (over 50 μM) of 6OHDA treatment leads to sustained ERK1/2 phosphorylation [44,47]. In our experiments, SHSY5Y cells were treated with high concentrations of 6OHDA and kept for 24 h, and the sustained ERK1/2 phosphorylation was decreased by amber treatment, which was consistent with the previous study [44].

In a previous study, α-synuclein aggregates were found to be a neuropathological hallmark of PD [5]. One of the important degradation pathways of α-synuclein is autophagic proteolysis (autophagy) [9,10]. Autophagy is an intracellular degradative process; it usually occurs under stress conditions, such as the presence of abnormal proteins and nutrient deficiency, etc. [48]. Rapamycin, an inducer of autophagy, has been shown to have a protective effect against PD-related dopaminergic neurodegeneration [10]. These findings show that autophagy may represent a potential neuroprotective strategy in PD. The results show that amber promotes autophagy, consistent with our previous study [21]. These results suggest that amber extract can protect neuronal cells against 6-OHDA-induced cell apoptosis by upregulating autophagy. Interestingly, an amber concentration of 50 µg/mL did not show the highest values of results in terms of autophagy, whereas a concentration of 25 µg/mL induced higher protein levels of both Beclin-1 and LC3II. However, in terms of the LC3 II/LC3 I ratio—a marker of autophagosome formation—the same effect was observed in the 25 µg/mL and 50 µg/mL groups. This indicates that a low concentration of amber extract may have a better therapeutic effect.

Neuroinflammation is also an important part of neurodegenerative diseases such as PD and AD [8,49,50]. In our study group, amber had an anti-inflammatory effect on the LPS-induced inflammatory cell model [18]. In previous studies, kujigamberol, which is isolated from Kuji amber, had anti-allergic effects [17,29]. Moreover, other bioactive compounds—such as monoterpenes, succinic monoterpenoids, and sesquiterpenoids—also have anti-inflammatory effects [24,25,27]. Thus, it is possible that the protective effects of amber extract may be the result of not only a single component, but also the combined action of many different types of components. In future research, we will focus on identifying these bioactive components in amber extract that can protect cells from 6-OHDA-induced cell death.

## 4. Materials and Methods

### 4.1. Materials

Dulbecco’s modified Eagle medium/nutrient mixture F-12 (DMEM/F12), 3-(4,5-dimethylthiazol-2-yl)-2,5-diphenylterazolium bromide (MTT), and 6-OHDA were purchased from Sigma-Aldrich (St. Louis, MO, USA). Fetal bovine serum (FBS) was purchased from HyClone Laboratories (Logan, UT, USA). Penicillin and streptomycin were purchased from Wako (Tokyo, Japan).

An apoptosis/necrosis detection kit (blue, green, red) and a DCFDA/H2DCFDA Cellular ROS Assay Kit were purchased from Abcam (Cambridge, UK). Antibodies such as Beclin, caspase-3, pERK, extracellular signal-regulated kinase (ERK), and LC3A/B XP, along with LumiGLO reagent, were purchased from Cell Signaling Technology (Danvers, MA, USA). RNAiso Plus was purchased from Takara Bio (Shiga, Japan). The THUNDERBIRD SYBR qPCR Mix was purchased from Toyobo (Tokyo, Japan).

### 4.2. Amber Extract

Baltic amber (Kaliningrad, Russia) was crushed, powdered, and extracted in 50% ethanol at 40 °C for 1 h under stirring, and then filtered. The extracted solution was depressurized and freeze-dried to form a powder (Kohaku Bio. Technology, Tsukuba, Japan). Amber extract powder was dissolved in dimethyl sulfoxide, and the mixture was stored at −80 °C until use.

### 4.3. Cell Culture

The human neuroblastoma cell line SHSY5Y (Riken Cell Bank, Tsukuba, Japan) was used for the experiments. SH-SY5Y cells were seeded in 96-well plates at 2 × 10^4^ cells/well and in 6 cm culture dishes at 1.3 × 10^6^ cells/dish for WB, cultured in DMEM/F12 medium with 10% FBS. After 24 h of seeding, the cells were pretreated with amber extract for 24 h, washed once with PBS, and then the medium was replaced with fresh 6OHDA-containing medium for another 24 h. The cells were maintained in a humidified atmosphere under 5% CO_2_ at 37 °C. The DMSO concentration in all experimental groups was unified to 0.05%.

### 4.4. MTT Assay

Cell viability was measured using the MTT assay. The medium was replaced with 90% DMEM/F12 and 10% MTT, and the cells were cultured at 37 °C for 4 h. Then, 10% sodium dodecyl sulfate (SDS) was added, and the mixture was kept at room temperature overnight. Absorbance was measured at 570 nm using a microplate reader (BioTek, Tokyo, Japan).

### 4.5. Apoptosis Assay

SHSY5Y cells were seeded in 96-well plates at 2 × 10^4^ cells/well, cultured in DMEM/F12 medium with 10% FBS. After 24 h of seeding, the cells were pretreated with amber extract for 24 h, followed by treatment with 6-OHDA for another 6 h. The apoptosis/necrosis detection kit (blue, green, red) was used to quantify apoptotic, necrotic, and live cells, according to the manufacturer’s protocol. Fluorescence was measured using a fluorescence microplate reader (BioTek, Tokyo, Japan), with fluorescence intensities of Ex/Em = 490/525 nm, Ex/Em = 550/650 nm, and Ex/Em = 405/450 nm for apoptotic, necrotic, and live cells, respectively.

### 4.6. ROS Assay

SHSY5Y cells were seeded in 96-well plates at 2 × 10^4^ cells/well, cultured in DMEM/F12 medium with 10% FBS. After 24 h of seeding, the cells were pretreated with amber extract for 24 h, followed by treatment with 6-OHDA for another 6 h. The DCFDA/H2DCFDA Cellular ROS Assay Kit was used to measure intracellular ROS concentration, according to the manufacturer’s protocol. Fluorescence was measured using a fluorescence microplate reader (BioTek, Tokyo, Japan), at a fluorescence intensity of Ex/Em = 485/535 nm.

### 4.7. Western Blotting

SHSY5Y cells were seeded in 6 cm culture dishes at 1.3 × 10^6^ cells/dish, cultured in DMEM/F12 medium with 10% FBS. After 24 h of seeding, the cells were pretreated with amber extract for 24 h, followed by treatment with 6-OHDA for another 24 h. The cells were collected via centrifugation, washed twice with cold phosphate-buffered saline, and lysed using RIPA buffer (150 mM NaCl, 1 mM EDTA, 50 mM Tris-HCl, 10 mM NaF, 1 mM Na_3_VO_4_, 1% Triton X-100, 0.1% SDS, 0.5% Na-deoxycholate, and protein inhibitor). After electrophoresis and transfer, the membrane was incubated with primary antibodies, including Beclin, caspase-3, pERK, ERK, and LC3 A/B XP. The membrane was then incubated with horseradish peroxidase (HRP)-conjugated secondary antibodies at room temperature for 1 h. LumiGLO reagent was used to detect HRP. The bands were detected using an AE-9300 Ez-Capture MG system (Atto Corporation, Tokyo, Japan). Protein expression levels were quantified using ImageJ software (NIH, Bethesda, MD, USA).

### 4.8. Data Analysis

All experiments were repeated at least thrice. The results are expressed as the mean ± SD. ANOVA was performed post hoc to compare data between the groups using SPSS Statistics (SPSS Inc., Chicago, IL, USA). Statistical significance was set at a *p*-value of < 0.05.

## 5. Conclusions 

Amber extract protected SHSY5Y cells from 6-OHDA-induced apoptosis. Additionally, it decreased intracellular ROS and phosphorylation of ERK, and promoted autophagy. These results suggest that amber can protect neuronal cells against 6-OHDA-induced cell death by upregulating autophagy and downregulating intracellular ROS. These results indicate that amber can potentially be used as a novel therapeutic and prophylactic candidate for PD.

## Figures and Tables

**Figure 1 molecules-27-01817-f001:**
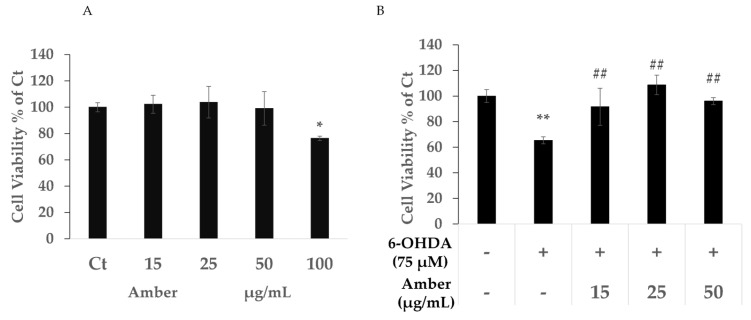
Effect of amber extract on 6-hydroxydopamine (6-OHDA)-induced cytotoxicity: SH-SY5Y cells were cultured in DMEM/F12 medium with 10% FBS. After 24 h of seeding, the cells were pretreated with amber extract for 24 h, and then the medium was replaced with fresh 6OHDA-containing medium for another 24 h, after which cell viability was measured using the MTT assay. Control (Ct): (6-OHDA/amber) −/−, 6-OHDA: (6-OHDA/amber) +/−, Amber: (6-OHDA/amber) (+/15 µg/mL; +/25 µg/mL, +/50 µg/mL). (**A**) Cytotoxicity of amber extract. (**B**) Effect of amber extract on 6-OHDA-induced cell death. Cell viability was determined using the 3-(4,5-dimethylthiazol-2-yl)-2,5-diphenylterazolium bromide assay. The results are expressed as the mean ± standard deviation. *n* ≥ 3, * *p* < 0.05 vs. Control, ** *p* < 0.01 vs. Control, ## *p* < 0.01 vs. 6-OHDA.

**Figure 2 molecules-27-01817-f002:**
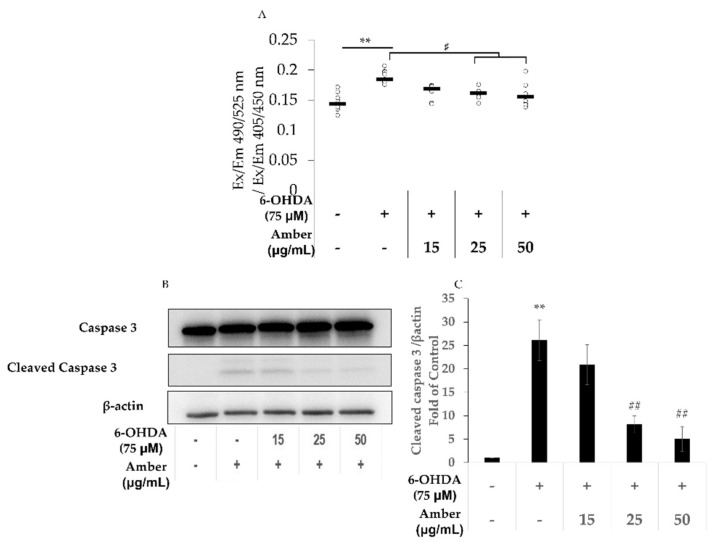
Effect of amber extract on 6-hydroxydopamine (6-OHDA)-induced cell apoptosis: SH-SY5Y cells were cultured in DMEM/F12 medium with 10% FBS. After 24 h of seeding, the cells were pretreated with amber extract for 24 h, and then the medium was replaced with fresh 6OHDA-containing medium for another 6 h for apoptosis assay, and 24 h for WB. Control (Ct): (6-OHDA/amber) −/−, 6-OHDA: (6-OHDA/amber) +/−, Amber: (6-OHDA/amber) (+/15 µg/mL; +/25 µg/mL, +/50 µg/mL). (**A**) Apoptosis analyzed using a fluorescence microplate reader. (**B**) Representative Western blot image of caspase-3 and cleaved caspase-3. (**C**) The protein bands of cleaved caspase-3 were quantified using ImageJ. The results are expressed as the mean ± standard deviation. *n* ≥ 3, ** *p* < 0.01 vs. Control, # *p* < 0.05 vs. 6-OHDA, ## *p* < 0.01 vs. 6-OHDA.

**Figure 3 molecules-27-01817-f003:**
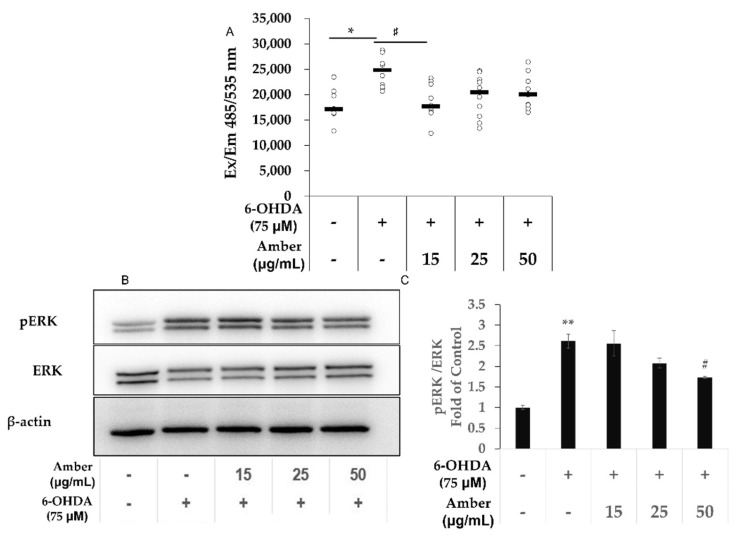
Effect of amber extract on 6-hydroxydopamine (6-OHDA)-induced intracellular reactive oxygen species (ROS) generation: SH-SY5Y cells were cultured in DMEM/F12 medium with 10% FBS. After 24 h of seeding, the cells were pretreated with amber extract for 24 h, and then the medium was replaced with fresh 6OHDA-containing medium for another 6 h for ROS assay, and 24 h for WB. Control (Ct): (6-OHDA/amber) −/−, 6-OHDA: (6-OHDA/amber) +/−, Amber: (6-OHDA/amber) (+/15 µg/mL; +/25 µg/mL, +/50 µg/mL) (**A**) ROS analyzed using a fluorescence microplate reader. (**B**) Representative Western blot image of pERK and extracellular signal-regulated kinase. (**C**) The protein bands were quantified using ImageJ. The results are expressed as the mean ± standard deviation. *n* ≥ 3 * *p* < 0.05 vs. Control, ** *p* < 0.01 vs. Control, # *p* < 0.05 vs. 6-OHDA.

**Figure 4 molecules-27-01817-f004:**
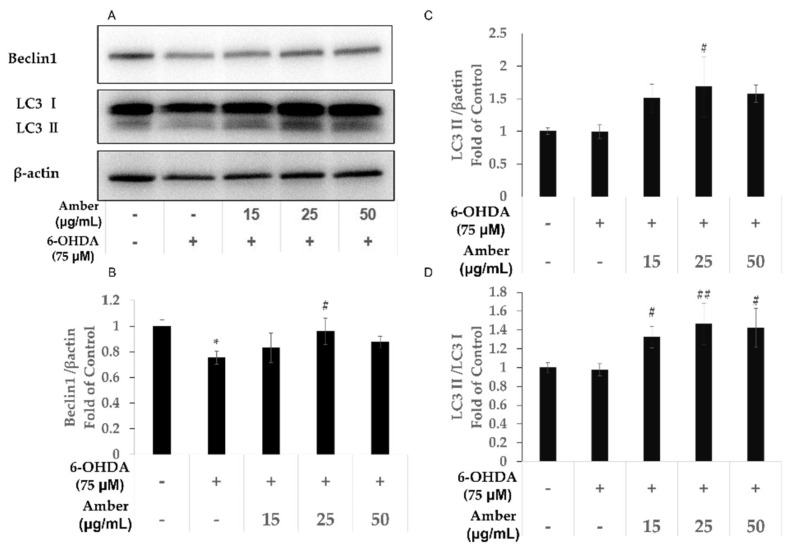
Effect of amber extract on autophagy-related gene expression: SH-SY5Y cells were cultured in DMEM/F12 medium with 10% FBS. After 24 h of seeding, the cells were pretreated with amber extract for 24 h, and then the medium was replaced with fresh 6OHDA-containing medium for another 24 h. Control (Ct): (6-OHDA/amber) −/−, 6-OHDA: (6-OHDA/amber) +/−, Amber: (6-OHDA/amber) (+/15 µg/mL; +/25 µg/mL, +/50 µg/mL) (**A**) Representative Western blot image of Beclin-1 and LC3. (**B**) The protein bands of Beclin-1 were quantified using ImageJ. (**C**) The protein bands of LC3 Ⅱ were quantified using ImageJ. (**D**) The relative ratios of LC3 II/LC3 I bands’ density. The results are expressed as the mean ± standard deviation. *n* ≥ 3 * *p* < 0.05 vs. Control, # *p* < 0.05 vs. 6-OHDA, ## *p* < 0.01 vs. 6-OHDA.

## Data Availability

All data are contained within the article.

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
