# Peer review of "Protective Effect of Amber Extract on Human Dopaminergic Cells against 6-Hydroxydopamine-Induced Neurotoxicity"

_molecules, 2022, doi:10.3390/molecules27061817_

Round 1

Reviewer 1 Report

The manuscript describes the beneficial properties of the amber extract on a PD  cellular model. The methodology is not clearly reported. Some information is missing that would make impossible reproducing the work. This is the main limitation of the manuscript. Figures should be improved and some of the original figures provided are duplicated.

Below you can find my specific comments.

Charts are difficult to read. the legend on the X axis is positioned in a weird way. It is not clear which compound is presents in a given experiment

More important, the actual values for each experiment should be reported. This way of representing the results is no more appropriate because they don't show the actual data. Please, see this paper: http://journals.plos.org/plosbiology/article?id=10.1371/journal.pbio.1002128 and adjust figures appropriately, showing the actual data.

In each graph the reference sample/experiment should be clearly depicted.

line 241 24h after what? How many cells were treated? What cellular density was achieved? Was the amber-containing medium removed before 6-OHDA treatment? Were the cells washed? How many times?  Were the cells detached before measuring apoptosis?
What was the control of the experiment? What about apoptosis in cells treated with amber extracts after 6 hours treatment?

Western blots. Were the WB performed on the same cells (i.e. a sub-population) used to measure apoptosis, viability, and ROS? or were independent cultures used? This would introduce additional experimental variability which should be taken into account.

Original figures. I see that the panels 2b (actin  caspase3 blot repeat 1) and 3b (actin ERK blot replicate 1) are identical. The same for replicate 2.  This is not acceptable.

Author Response

Reviewer #1

Responses to the Comments by the Reviewer #1

The manuscript describes the beneficial properties of the amber extract on a PD  cellular model. The methodology is not clearly reported. Some information is missing that would make impossible reproducing the work. This is the main limitation of the manuscript. Figures should be improved and some of the original figures provided are duplicated.

Reply:

Thank you for your comments.

Below you can find my specific comments.

Charts are difficult to read. the legend on the X axis is positioned in a weird way. It is not clear which compound is presents in a given experiment

More important, the actual values for each experiment should be reported. This way of representing the results is no more appropriate because they don't show the actual data. Please, see this paper: http://journals.plos.org/plosbiology/article?id=10.1371/journal.pbio.1002128 and adjust figures appropriately, showing the actual data.

In each graph the reference sample/experiment should be clearly depicted.

Reply:

Thank you for your comment.

We revised the description of X axis to make it easier to be understanded. Please refer to Fig.1-4.

Also, we revised the figures of ROS and apoptosis results to let the data be showed more appropriate. the Fig.2 and 3.

We revised the reference of graph more clearly, please refer to Fig.1-4.

.
line 241 24h after what? How many cells were treated? What cellular density was achieved? Was the amber-containing medium removed before 6-OHDA treatment? Were the cells washed? How many times?  Were the cells detached before measuring apoptosis?
What was the control of the experiment? What about apoptosis in cells treated with amber extracts after 6 hours treatment?

Reply:

Thank you for your comment. We revised the depicted of method in cell culture, please refer to method section, line263-268.

For the apoptosis experiment, because the cells are adherent cell, therefore, we did not detached cells. The cells were directly treated by reagent in 96-well-plates, and the fluorescence was measured using a fluorescence microplate reader.

The control group for this experiment was the group without 6OHDA and amber. Because amber treatment did not show a decrease in cell viability, so, we think it is not necessary to do the apoptosis, so in this experiment we did not measure the effect of amber on apoptosis in this experiment.

Open Review

Western blots. Were the WB performed on the same cells (i.e. a sub-population) used to measure apoptosis, viability, and ROS? or were independent cultures used? This would introduce additional experimental variability which should be taken into account.

Reply:

Thank you for your comment. WB experiments were used the same cell density and the same cell line. And all experiments were repeated at least three independent replicates.

Original figures. I see that the panels 2b (actin  caspase3 blot repeat 1) and 3b (actin ERK blot replicate 1) are identical. The same for replicate 2.  This is not acceptable.

Reply:

Thank you for your comment. We used the same membrane to do WB for several different kind of protein, like ERK and Caspase 3 and so on, and after all protein measured, we measured the β-actin. This is the reason of some results have the same β-actin band.  

Reviewer 2 Report

The study shows the potential use of amber extract as a neuroprotective agent in a cell-based model. In general, the paper is well-written, but it lacks some information on the experimental design and extract origin. Further clarification is needed. 

  1. Is the raw material (amber) commercially available (Line 285-286)? If yes, please provide the details of the product (product number, company, country of origin) in the 'Amber extract' section. If possible, please provide the product picture as well. 
  2. Line 112-137, the authors explain the biological activities of terpenoids which is too brief and not specific. Since the author did not perform any chemical profiling of the extract (which is important for plant extract authentication), please discuss specify what are the reported compounds in the Baltic amber and their neuroprotective activities. 
  3. Please specify the final percentage of DMSO (as the negative control) in the cell-based studies. This is important because the SHSY5Y cells are highly sensitive to DMSO. 
  4. Differentiated SHSY5Y cells tend to be more neuronal-like. But the authors did not use a differentiated SHSY5Y cells model in this study. Please justify and include either in introduction or discussion part. 

Author Response

Reviewer #2

Responses to the Comments by the Reviewer #2

The study shows the potential use of amber extract as a neuroprotective agent in a cell-based model. In general, the paper is well-written, but it lacks some information on the experimental design and extract origin. Further clarification is needed. 

Reply:

Thank you for your comments.

  1. Is the raw material (amber) commercially available (Line 285-286)? If yes, please provide the details of the product (product number, company, country of origin) in the 'Amber extract' section. If possible, please provide the product picture as well. 

Reply:

Thank you for your comment. The amber used in this experiment is custom-made products from Kaliningrad, Russia,imported from esprima company (Oosaka, Japan). It is not commercially available.

  1. Line 112-137, the authors explain the biological activities of terpenoids which is too brief and not specific. Since the author did not perform any chemical profiling of the extract (which is important for plant extract authentication), please discuss specify what are the reported compounds in the Baltic amber and their neuroprotective activities. 

Reply:

Thank you for your comment. The purification and composition analysis of amber is carried out by the other group in our lab. So, we added assumptions about the active ingredients in amber, this was also added in discussion section, Line147-151.

  1. Please specify the final percentage of DMSO (as the negative control) in the cell-based studies. This is important because the SHSY5Y cells are highly sensitive to DMSO. 

Reply:

Thank you for your comment. Adding different concentrations of amber extract to the medium will result in inconsistent DMSO concentration. Therefore, in this experiment, the DMSO concentration in all experimental groups in experiments was unified to 0.05%. Thank you very much for your suggestion to help us improve our paper. The revision of this part, we have added to methods section, Line268-269. 

  1. Differentiated SHSY5Y cells tend to be more neuronal-like. But the authors did not use a differentiated SHSY5Y cells model in this study. Please justify and include either in introduction or discussion part. 

Reply:

Thank you for your comment.

Differentiated SHSY5Y cells are more neuronal-like, however, in previous study, differentiated SHSY5Y cells have some alteration in the AKT pathway and have higher tolerance to the toxicity of 6-OHDA, which means, undifferentiated SHSY5Y cells are more suitable in Parkinson's Disease model. Therefore, undifferentiated SHSY5Y cells were used in this study. The reason for using undifferentiated SHSY5Y cells was also added into the discussion section, Line165-168.

Reviewer 3 Report

Article entitled "Protect Effect of Amber Extract on Human Dopaminergic Cells Against 6-hydroxydopamine-induced Neurotoxicity" by Luo et al. is interesting and raises a fairly new topic of molecular research on the effect of amber extract on cells and the potential use of this extract in medicine. In my opinion, authors should revise the article to be reconsidered for publication. Below I am sending my comments and remarks that will improve the manuscript.

Comments:

  • line 17 – sentence „In addition, we determined its potential effect on PD” should be explained more, how was this examined?
  • it should be mentioned in the abstract that the SHSY5Y cells tested are human neuroblastoma cell line
  • line 19/20 – “amber extract decreases apoptosis” – the sentence should be rewritten e.g. to “amber extract reduces apoptosis” or “amber extract has a protective effect against apoptosis”
  • line 32 – should be 2–4% instead of 2%–4%
  • line 32 and others – citations throughout the text should be changed to [2,3] instead of [2][3]
  • line 45 – “in vitro” should be in italics
  • there are double spaces or no spaces in the text
  • line 59/60 – “evaluate its potential in PD treatment” – I believe that this sentence should be removed as it suggests animal or human research
  • figures are of poor quality
  • the description of figures in the text should be standardized – sometimes they are described with a capital letter and sometimes with a lower case letter (lines 97 and 110)
  • lines 195 and 195 – discussion should not contain references to the figures
  • line 265 – a detailed description of the statistical software used should be added (name, company, country)
  • references should be corrected and adjusted to the journal's requirements
  • a good supplement to the article would be a chromatographic analysis of the analyzed amber extract
  • article can be supplemented with tests of the antioxidant capacity of extracts in in vitro tests, e.g. using the ABTS method
  • is anything known about the toxicity of amber extract from other studies? authors should comment on this fact in discussion
  • conclusions need to be expanded

Author Response

Reviewer #3

Responses to the Comments by the Reviewer #3

Article entitled "Protect Effect of Amber Extract on Human Dopaminergic Cells Against 6-hydroxydopamine-induced Neurotoxicity" by Luo et al. is interesting and raises a fairly new topic of molecular research on the effect of amber extract on cells and the potential use of this extract in medicine. In my opinion, authors should revise the article to be reconsidered for publication. Below I am sending my comments and remarks that will improve the manuscript.

Reply:

Thank you for your comments.

  • line 17 – sentence „In addition, we determined its potential effect on PD” should be explained more, how was this examined?

Reply:

Thank you for your comment. As your advice, we also found this discription was not accurate here. Therefore, we revised the sentences from “In addition, we determined its potential effect on PD” to “Initially, we determined the protective effect of amber on PD model.”, please refer to line 16.

  • it should be mentioned in the abstract that the SHSY5Y cells tested are human neuroblastoma cell line

Reply:

Thank you for your comment. We revised it, please refer line 15.

  • line 19/20 – “amber extract decreases apoptosis” – the sentence should be rewritten e.g. to “amber extract reduces apoptosis” or “amber extract has a protective effect against apoptosis”

Reply:

Thank you for your comment. We revised it, please refer line 17.

  • line 32 – should be 2–4% instead of 2%–4%

Reply:

Thank you for your comment. We revised it, please refer line 32.

  • line 32 and others – citations throughout the text should be changed to [2,3] instead of [2][3]

Reply:

Thank you for your comment. We are so sorry for our mistake. We revised all the mistake on citations.

  • line 45 – “in vitro” should be in italics

Reply:

Thank you for your comment. We revised it, please refer line 45

  • there are double spaces or no spaces in the text

Reply:

Thank you for your comment. We are so sorry for our mistake. We checked again and revised the mistake.

  • line 59/60 – “evaluate its potential in PD treatment” – I believe that this sentence should be removed as it suggests animal or human research

Reply:

Thank you for your comment. We found this sentence was not suitable because we don’t have the animal/ human research, Therefore, we removed the sentences “evaluate its potential in PD treatment”, please refer to line 59/60.

  • figures are of poor quality

Reply:

Thank you for your comment. The resolution of all Figures has been increased to 350 dpi.

  • the description of figures in the text should be standardized – sometimes they are described with a capital letter and sometimes with a lower case letter (lines 97 and 110)

Reply:

Thank you for your comment. We are so sorry for our mistake. We checked again and revised the mistake.

  • lines 195 and 195 – discussion should not contain references to the figures

Reply:

Thank you for your comment. We revised it from “Beclin1 and LC3II (Fig.4b&4c)”; “50 µg/mL group (Fig.4d).to “Beclin1 and LC3II”; “50 µg/mL group”. Please refer to line 226,228.

  • line 265 – a detailed description of the statistical software used should be added (name, company, country)

Reply:

Thank you for your comment. We revised it, please refer line313-314.

  • references should be corrected and adjusted to the journal's requirements

Reply:

Thank you for your comment. We are so sorry for our mistake. We checked again and revised it.

  • a good supplement to the article would be a chromatographic analysis of the analyzed amber extract

Reply:

Thank you for your comment. The purification and composition analysis of amber is carried out by the other group in our lab. Therefore, we added assumptions about the active ingredients in amber, this was also added in discussion in Line147-151.

  • article can be supplemented with tests of the antioxidant capacity of extracts in in vitrotests, e.g. using the ABTS method

Reply:

Thank you for your comment. we also test the antioxidant capacity of amber solution by DPPH assay. The results showed that high concentration of amber show a little ability on antioxidant, but the concentration used in the experiment(~50μg/ml) didn’t showed antioxidant.

The result is like below.

  • is anything known about the toxicity of amber extract from other studies? authors should comment on this fact in discussion

Reply:

Thank you for your comment. About amber extract, there is still don’t have much evidence yet. In our study group, we found that low concentration of amber showed no toxicity in vitro and vivo.  

  • conclusions need to be expanded

Reply:

Thank you for your comment. We revised the conclusion, please refer line320-321.

Author Response

Reviewer #4

Responses to the Comments by the Reviewer #4

 This manuscript describes novel protective effect of amber extract on neurotoxicity caused by 6-hydroxydopamine. The authors have demonstrated that treatment with amber extract decreased 6-hydroxydoapmine induced apoptosis, decreased ROS, and promoted autophagy. The likely cellular candidates were shown to be related to ERK signaling pathway. The study proposes potential use of amber as a novel therapeutic for PD. Most of the conclusions are based on the experimental data provided in the manuscript. However, there are several major points that need to be addressed to strengthen the claims made by this study.

Reply:

Thank you for your comments.

  1. The mechanistic studies conducted showed that various doses of amber extract were able to decrease p-ERK levels in SH-SY5Y cells at one-time point. However, earlier studies have indicated that early activation of ERK is a protective mechanism exhibited by dopaminergic cells (PMID: 17847117). Thus inhibition of the early rise of p-ERK leads to increase in ROS levels. Although authors have demonstrated beneficial effect of amber on ROS, it is important to look at the effect of amber treatment on p-ERK and ROS levels at an early time point or a time course would be appropriate.

Reply:

Thank you for your comment. We gratefully appreciate for your valuable suggestion, as the pervious study showed short time treatment of 6OHDA can lead to distinct temporal pattern of the activation of ERK, and early activation of ERK is a protective mechanism, and the inhibitions of the first activation leads to the increase in ROS. However, in our study, we collected and measured the protein at 24h after treated with 6OHDA, and a decrease in pERK/ERK was found at this time point.

Previous studies also showed that high concentration and long-time treatment of 6OHDA is relative to sustained ERK1/2 phosphorylation, which finally cause neuronal cell death(Park et al. 2013)(Kulich et al. 2001).

In this experiment, we treated the cell with a high concentration and kept for 24h, which may also have induced a sustained ERK1/2 phosphorylation. This was also added in discussion in Line207-214.

  1. How are the antioxidant levels affected upon amber treatment?

Reply:

Thank you for your comment. In our pervious study, amber extract can upregulate the mRNA level of SOD1 and SOD2 to downregulate ROS generation. However, according to the DPPH assay results, the concentration used in this experiment have no antioxidant capacity. Therefore, decreased of ROS may be relative with the SOD1 and SOD2.

  1. One of the mechanism of 6-hydroxydopamine toxicity is its effect on mitochondrial function. Studies directed at the effect of amber on mitochondrial redox activity and potential could substantiate its potential as a potential PD therapeutic.

Reply:

Thank you for your comment. Mitochondrial dysfunctions play an important role in PD. We will do the future study on it after the composition analysis. Therefore, we added assumptions about the active ingredients may have the effect on mitochondrial function in amber, this was also added in discussion in Line183-187.

  1. Procedure is provided for preparation of the amber extract. Some discussion on availability of amber and reproducibility in preparation of amber extract could be useful.

Reply:

Thank you for your comment. The purification and composition analysis of amber is carried out by the other group in our lab. As so far, we had prepared amber extract in several times, and all trial show the similar tendency on composition.

  1. Statistical analysis is conducted using the student’s t-test. Comparison of multiple groups using this test could be lead to type-I error. It is essential to provide statistical analysis using appropriate ANOVA test.

Reply:

Thank you for your comment. In this revision, we analyzed the data by ANOVA post-hoc test.

Minor points:

  1. Manuscript needs extensive grammatical and language corrections. The title should also state “Protective effect of amber….”.

Reply:

Thank you for your comment. We check the grammatical and language again and revised it. 

  1. Referring to SH-SY5Y as dopaminergic cells is misleading. It should be stated as a dopaminergic cell model.

Reply:

Thank you for your comment. We are so sorry for our mistake. We checked and revised it.

  1. Some discussion of the role of ERK pathway in 6-hydroxydopamine-induced ROS should be included.

Reply:

Thank you for your comment. We gratefully appreciate for your valuable suggestion. This was added in discussion in Line198-199, 203-205.

Round 2

Reviewer 1 Report

I wish to thank the authors for responding to my questions and for revising the manuscript according to my comments and suggestions.

I have still some conncerns about the quality of figures and the statistical evaluation of the results obtained.

Figure 1
in panel B the concentration of 6-OHDA used should be inndicated innstead of "+". Thi sfacilitate the readers. I also suggest to change the position of "ug/mL", placing it in line with the X axis or in line with 6-OHDA and Amber 

Also in panel B the p value above the third bar has changed compared to the previous version of the manuscript. Is this correct? Can the authors provide the raw data for the experiments depicted in Figure 1 (for both panels)?

Similarly, in figures 2 and 3 some condition have gained significance while other have lost. Is there an explanation for those changes?  I recommend to check carefully how the statistical tests have been performed before publication. Again, publishing the raw results as supplementary information would be the best option.

Author Response

Reviewer #1

Responses to the Comments by the Reviewer #1

I wish to thank the authors for responding to my questions and for revising the manuscript according to my comments and suggestions.

Reply:

Thank you for your comment.

I have still some conncerns about the quality of figures and the statistical evaluation of the results obtained.

Figure 1
in panel B the concentration of 6-OHDA used should be inndicated innstead of "+". Thi sfacilitate the readers. I also suggest to change the position of "ug/mL", placing it in line with the X axis or in line with 6-OHDA and Amber 

Reply:

Thank you for your comment. We revised it, please refer Fig1-4.

Also in panel B the p value above the third bar has changed compared to the previous version of the manuscript. Is this correct? Can the authors provide the raw data for the experiments depicted in Figure 1 (for both panels)?

Similarly, in figures 2 and 3 some condition have gained significance while other have lost. Is there an explanation for those changes?  I recommend to check carefully how the statistical tests have been performed before publication. Again, publishing the raw results as supplementary information would be the best option.

Reply:

Thank you for your comment.
About the p value, in the first version we used student’s t test to analysis the data, and as reviewer #4’s advice, we re-analyzed our data with ANOVA post-hoc test, which is the reason that the p value in revision was different with the previous version.
As your advice, we changed the relative value into actual data (raw data) in Fig 2a ROS and Fig 3a apoptosis’s results, and we checked the statistical tests again.  

In addition, if you still think it is necessary to check the raw data, we will send it to you.

Reviewer 2 Report

The authors have answered most of the comments. The article can be considered for acceptance now. TQ

Author Response

Reviewer #2

Responses to the Comments by the Reviewer #2

The authors have answered most of the comments. The article can be considered for acceptance now. TQ

Reply:

Thank you very much.

Reviewer 3 Report

The authors corrected the article according to my comments and responded to all remarks.

Before publication, I propose to improve the quality of the figures - in some places the description of the axis is blurred. References still need to be formatted - the year of publication should be in bold and page numbers should not be in italics.

Author Response

Reviewer #3

Responses to the Comments by the Reviewer #3

The authors corrected the article according to my comments and responded to all remarks.

Reply:

Thank you for your comment.

Before publication, I propose to improve the quality of the figures - in some places the description of the axis is blurred.

Reply:

Thank you for your comment. we changed the size of texts in figures, and the resolution of all Figures has been increased to 500 dpi. Please refer to Fig.1-4.

References still need to be formatted - the year of publication should be in bold and page numbers should not be in italics.

Reply:

Thank you for your comment. We are so sorry for our mistake. We checked again and revised it.

Reviewer 4 Report

The authors have addresses majority of the concerns raised by this reviewer . The current version of the manuscript is suitable for publication.

This manuscript describes novel protective effect of amber extract on neurotoxicity caused by 6- hydroxydopamine. The authors have demonstrated that treatment with amber extract decreased 6-hydroxydoapmine induced apoptosis, decreased ROS, and promoted autophagy. The likely cellular candidates were shown to be related to ERK signaling pathway. The study proposes potential use of amber as a novel therapeutic for PD. Most of the conclusions are based on the experimental data provided in the manuscript. However, there are several major points that need to be addressed to strengthen the claims made by this study.
1. The mechanistic studies conducted showed that various doses of amber extract were able to decrease p-ERK levels in SH-SY5Y cells at one-time point. However, earlier studies have indicated that early activation of ERK is a protective mechanism exhibited by dopaminergic cells (PMID: 17847117). Thus inhibition of the early rise of p-ERK leads to increase in ROS levels. Although authors have demonstrated beneficial effect of amber on ROS, it is important to look at the effect of amber treatment on p-ERK and ROS levels at an early time point or a time course would be appropriate.
2. How are the antioxidant levels affected upon amber treatment?
3. One of the mechanism of 6-hydroxydopamine toxicity is its effect on mitochondrial function. Studies directed at the effect of amber on mitochondrial redox activity and potential could substantiate its potential as a potential PD therapeutic.
4. Procedure is provided for preparation of the amber extract. Some discussion on availability of amber and reproducibility in preparation of amber extract could be useful.
5. Statistical analysis is conducted using the student’s t-test. Comparison of multiple
groups using this test could be lead to type-I error. It is essential to provide statistical analysis using appropriate ANOVA test.

Minor points:
6. Manuscript needs extensive grammatical and language corrections. The title should also state “Protective effect of amber….”.
7. Referring to SH-SY5Y as dopaminergic cells is misleading. It should be stated as a dopaminergic cell model.
8. Some discussion of the role of ERK pathway in 6-hydroxydopamine-induced ROS should be included.

Author Response

Reviewer #4

Responses to the Comments by the Reviewer #4

The authors have addresses majority of the concerns raised by this reviewer . The current version of the manuscript is suitable for publication.

Reply:

Thank you very much.

I have already answered the below mentioned comments in revision 1. However, I have again mentioned the same answers below. Thank you very much.

This manuscript describes novel protective effect of amber extract on neurotoxicity caused by 6- hydroxydopamine. The authors have demonstrated that treatment with amber extract decreased 6-hydroxydoapmine induced apoptosis, decreased ROS, and promoted autophagy. The likely cellular candidates were shown to be related to ERK signaling pathway. The study proposes potential use of amber as a novel therapeutic for PD. Most of the conclusions are based on the experimental data provided in the manuscript. However, there are several major points that need to be addressed to strengthen the claims made by this study.

Reply:

Thank you for your comment.

  1. The mechanistic studies conducted showed that various doses of amber extract were able to decrease p-ERK levels in SH-SY5Y cells at one-time point. However, earlier studies have indicated that early activation of ERK is a protective mechanism exhibited by dopaminergic cells (PMID: 17847117). Thus inhibition of the early rise of p-ERK leads to increase in ROS levels. Although authors have demonstrated beneficial effect of amber on ROS, it is important to look at the effect of amber treatment on p-ERK and ROS levels at an early time point or a time course would be appropriate.

Reply:

Thank you for your comment. We gratefully appreciate for your valuable suggestion, as the pervious study showed short time treatment of 6OHDA can lead to distinct temporal pattern of the activation of ERK, and early activation of ERK is a protective mechanism, and the inhibitions of the first activation leads to the increase in ROS. However, in our study, we collected and measured the protein at 24h after treated with 6OHDA, and a decrease in pERK/ERK was found at this time point.

Previous studies also showed that high concentration and long-time treatment of 6OHDA is relative to sustained ERK1/2 phosphorylation, which finally cause neuronal cell death(Park et al. 2013)(Kulich et al. 2001).

In this experiment, we treated the cell with a high concentration and kept for 24h, which may also have induced a sustained ERK1/2 phosphorylation. This was also added in discussion in Line207-214.

  1. How are the antioxidant levels affected upon amber treatment?

Reply:

Thank you for your comment. In our pervious study, amber extract can upregulate the mRNA level of SOD1 and SOD2 to downregulate ROS generation. However, according to the DPPH assay results, the concentration used in this experiment have no antioxidant capacity. Therefore, decreased of ROS may be relative with the SOD1 and SOD2.

  1. One of the mechanism of 6-hydroxydopamine toxicity is its effect on mitochondrial function. Studies directed at the effect of amber on mitochondrial redox activity and potential could substantiate its potential as a potential PD therapeutic.

Reply:

Thank you for your comment. Mitochondrial dysfunctions play an important role in PD. We will do the future study on it after thecomposition analysis. Therefore, we added assumptions about the active ingredients may have the effect on mitochondrial function in amber, this was also added in discussion in Line183-187.

  1. Procedure is provided for preparation of the amber extract. Some discussion on availability of amber and reproducibility in preparation of amber extract could be useful.

Reply:

Thank you for your comment. The purification and composition analysis of amber is carried out by the other group in our lab. As so far, we had prepared amber extract in several times, and all trial show the similar tendency on composition.

  1. Statistical analysis is conducted using the student’s t-test. Comparison of multiple groups using this test could be lead to type-I error. It is essential to provide statistical analysis using appropriate ANOVA test.

Reply:

Thank you for your comment. In this revision, we analyzed the data by ANOVA post-hoc test.

Minor points:

  1. Manuscript needs extensive grammatical and language corrections. The title should also state “Protective effect of amber….”.

Reply:

Thank you for your comment. We check the grammatical and language again and revised it. 

  1. Referring to SH-SY5Y as dopaminergic cells is misleading. It should be stated as a dopaminergic cell model.

Reply:

Thank you for your comment. We are so sorry for our mistake. We checked and revised it.

  1. Some discussion of the role of ERK pathway in 6-hydroxydopamine-induced ROS should be included.

Reply:

Thank you for your comment. We gratefully appreciate for your valuable suggestion. This was added in discussion in Line198-199, 203-205.
